# *Xenopus laevis* (Daudin, 1802) as a Model Organism for Bioscience: A Historic Review and Perspective

**DOI:** 10.3390/biology12060890

**Published:** 2023-06-20

**Authors:** Rosa Carotenuto, Maria Michela Pallotta, Margherita Tussellino, Chiara Fogliano

**Affiliations:** 1Department of Biology, University of Naples Federico II, 80126 Naples, Italy; tussemar@libero.it; 2Institute for Biomedical Technologies, National Research Council, 56124 Pisa, Italy; mariamichela.pallotta@irgb.cnr.it

**Keywords:** model organism, *Xenopus laevis*, biological mechanisms

## Abstract

**Simple Summary:**

*Xenopus laevis* is an anuran amphibian that has been used for decades as the principal vertebrate model to study embryonic development. Among the advantages of this species, one can mention its evolutionary closeness to higher vertebrates in terms of physiology, gene expression and organ development. In recent years, this model has gradually attracted the attention of ecotoxicologists, since it can be a good indicator of habitat diversity, biological variety and local stressors. In this review, we summarize the main available literature on the use of *X. laevis* in different biological fields, highlighting its advantages and great versatility.

**Abstract:**

In vitro systems have been mainly promoted by authorities to sustain research by following the 3Rs principle, but continuously increasing amounts of evidence point out that in vivo experimentation is also of extreme relevance. *Xenopus laevis*, an anuran amphibian, is a significant model organism in the study of evolutionary developmental biology, toxicology, ethology, neurobiology, endocrinology, immunology and tumor biology; thanks to the recent development of genome editing, it has also acquired a relevant position in the field of genetics. For these reasons, *X. laevis* appears to be a powerful and alternative model to the zebrafish for environmental and biomedical studies. Its life cycle, as well as the possibility to obtain gametes from adults during the whole year and embryos by in vitro fertilization, allows experimental studies of several biological endpoints, such as gametogenesis, embryogenesis, larval growth, metamorphosis and, of course, the young and adult stages. Moreover, with respect to alternative invertebrate and even vertebrate animal models, the *X. laevis* genome displays a higher degree of similarity with that of mammals. Here, we have reviewed the main available literature on the use of *X. laevis* in the biosciences and, inspired by Feymann’s revised view, “Plenty of room for biology at the bottom”, suggest that *X. laevis* is a very useful model for all possible studies.

## 1. Introduction

*Xenopus laevis* (Daudin, 1802), also known as the African clawed frog, is a tetraploid species that has been used as one of the principal vertebrate models since the 1950s [1,2,3]. Over the years, it has been widely used to better understand a broad range of mechanisms related to embryogenesis and organogenesis, toxicology and also human disease mechanisms. This is because this anuran amphibian produces a great number of eggs (approximately one hundred) that are easily fertilized both in vitro and in vivo and that, therefore, can produce a countless number of embryos in a few days, during all seasons [4,5]. The embryonic stages have been frequently employed because of their advantages in terms of time, cost and ease of handling, thus allowing easy manipulation, with the possibility of producing molecular and mechanical alterations to study the restoration of normal conditions and the pathways involved. From an ethical point of view, the use of the *X. laevis* embryo is not debatable because, as with zebrafish larvae, early *Xenopus* development falls outside the 240/2010 EU directive. Indeed, until stage 46/47 (96–120 h pf at 22/23 °C according to Nieuwkoop and Faber [6]), they are not actively feeding, because they are still using the remaining yolk. This means that no approval from the health authorities is needed for in vivo experiments that involve the early stages of this species. Among the great advantages of *Xenopus*, there is its evolutionary closeness to higher vertebrates in terms of physiology, gene expression and organ development, such as the eyes, liver, lungs, heart, kidney and so on, thus allowing the transfer of the results directly to the human species [7,8,9,10,11,12]. In recent years, this model of animal species has gradually attracted the attention of ecotoxicologists and been applied in studies relating to ecotoxicology. It can be, in fact, a good indicator of habitat diversity, biological variety and local stressors, in both water and air, due to its external fertilization and development and permeable skin, which is susceptible to environmental pollutants due to direct exposure to the environment. Thus, it is a good reflection of environmental pollution [2,13,14]. Because of its importance as a model organism for studies on developmental biology, *X. laevis* has been chosen as one of the main species for in vitro teratogenesis studies through the Frog Embryo Teratogenesis Assay—*Xenopus* (FETAX) test, a specific assay that uses early-stage embryos of this species to measure the activity of pollutants during development (Figure 1).

The importance of the use of *Xenopus* as a model organism, both adults and embryos, is evidenced by the very large number of papers published in all areas of bioscience research (circa 10,000 as of 2008 to the present, according to PubMed). With the present review, our aim is to contribute by highlighting some particular features that could make the amphibian *X. laevis* a suitable model for several studies, aiding in our understanding of the composite mechanisms of more complex organisms within toxicology and ecotoxicology studies.

## 2. The Advantages of the *Xenopus* Model

Among the disadvantages of the use of amphibians in biology research, the fact that they are seasonal breeders would seem to be one of the greatest limitations, considering the impossibility to conduct experiments throughout the year. In this scenario, *X. laevis* is a notable exception. In fact, with an injection of chorionic gonadotropin (contained in pregnancy urine), it is easy to induce spawning and, therefore, to overcome the limitations imposed by the natural reproduction period. It was this possibility that led to its common usage for human pregnancy tests in the 1950s. For this reason, investigators considered its use in experimental embryology as early as the first half of the last century.

There are several advantages of its usage in research areas:It is easy to obtain from dedicated companies and to raise and feed in large colonies in the laboratory;It is possible to induce, in the female, repeated spawning at any season, and therefore to obtain embryos throughout the year;Thanks to their relatively large dimensions (~1.0–1.3 mm in diameter), it is possible to easily handle the oocytes before the fertilization process; moreover, oocytes are released in large numbers, making in vitro fertilization possible;Since both fertilization and embryonic development take place outside the mother’s body, it is possible to observe and manipulate the process at every stage (Figure 2);Its early development is very fast, making it possible to study the developmental progress, until the tadpole stage, in approximately four days at 22/24 °C;It is an excellent model for the study of body axis formation;It has a high degree of conservation of most essential cellular and molecular mechanisms.

## 3. History of *Xenopus* Use in Laboratory

The use of amphibians in biological studies dates back to 1958 and is evidenced by published laboratory textbooks in which it is described the successful handling of these animals [16,17]. As assessed in Gurdon and Hopwood’s review [1], the introduction of *Xenopus* into developmental biology laboratories was rather fortuitous. In their publication, the reader is directed to a survey of these authors that analyzes in detail how the South African *Xenopus* won a never-announced competition over an amphibian genus widely used in Europe (*Triturus, Bufo, Discoglossus*) and the USA (*Rana pipiens*). Following this, it was reported how *X. laevis* can be easily bred in captivity and that the standard protocols are simpler and of a lower cost than those required for rodents, constituting important reasons that make *Xenopus* such a successful animal model [18]. Besides the main advantages constituted by its embryonic development outside the maternal body, the *Xenopus* genus has some disadvantages, such as the long generation time and the allotetraploid genome, features that make genetics studies more difficult to perform. Despite a few disadvantages, over time, the use of this amphibian in the laboratory was further encouraged by the following advantages that it offers. These include (1) the opportunity to easily manipulate gene expression through the simple microinjection of constructs into the oocyte or in two-cell embryos [19]; (2) the possibility of following organogenesis without resorting to invasive approaches [20,21]; and (3) the availability of a fully sequenced genome (for both *X. laevis* and *X. tropicalis*) [10,22]. Amphibians, the *Xenopus* species included, diverged more recently from amniotes (360 million years ago) than fish (over 400 million years ago), and their complete genome sequencing revealed a high degree of synteny with humans, with 90% of human disease gene homologs [10]. Another advantage is (4) the availability of *Xenopus*’ genetic and genomic data in order to study genes, gene families and gene networks, including expressed sequence tags (ESTs) [23,24], as well as (5) the continuous update of UniGene clusters and the NCBI database of the transcriptome, with new genomic sequences for both *X. laevis* and *X. tropicalis*. This allows the use of high-throughput technologies in *Xenopus*, including RNA-Seq [25] and quantitative proteomics [26]. Furthermore, the ability to access an increasing resource of transgenic lines owned by the National *Xenopus* Resource (NXR), the European *Xenopus* Resource Centre (EXRC) and other stock centers [27] is an advantage, as well as (6) the opportunity to use techniques for gene expression modification (for example, antisense morpholino oligonucleotides) to generate transgenic animals, and to utilize genome editing techniques such as the CRISPR/Cas9 system [28,29]. Moreover, since many essential processes at the cellular and molecular level are highly conserved across vertebrates, *X. laevis* can be considered a valuable organism to study the mechanisms of human disease, cancer above all. The use of the *Xenopus* model system has provided, over the years, several contributions to cancer research, mainly given by the striking parallels between tumor pathogenesis and *Xenopus*’ early embryo development pathways [30,31,32,33,34].

## 4. *Xenopus* in Developmental Studies

### 4.1. Xenopus in the Study of Oogenesis and Role of Cytoskeletal Proteins

During meiotic maturation, oocytes are transcriptionally repressed, and all necessary proteins are translated from preexisting maternally derived mRNAs. This specific property led scientists to consider injecting poly(A) + -mRNA, extracted from mammalian tissues, into *Xenopus* oocytes. The result was that the corresponding protein was induced, indicating that the translation machinery was active for foreign messengers. The oocytes were then utilized in a laboratory to produce proteins at a time when other tools to obtain large amounts of proteins were still unknown [35,36,37,38]. When sufficient attention was paid to the oocytes themselves, the morphology and the functions of the oocytes were analyzed in several studies, including that of the cytoskeleton. The occurrence of the spatial distribution of organelles and molecules, pivotal for oocyte organization, is strongly dependent upon the cytoskeleton and the viscoelastic conditions of the cytoplasm. A unique stage-dependent cytoskeletal organization, if compared to oocytes of other amphibians (*Discoglossus*, *Bufo*, *Rana*, etc.), is present in the late stages of *Xenopus*’ oogenesis. In previtellogenic oocytes, the CSK forms a robust network in the cortex that gradually spreads throughout the cell. Here, it is possible to find tubulin and an actin-based cytoskeleton, including vinculin, vimentin, talin, cytokeratin, spectrin and myosin (for a review, see Carotenuto and Tussellino [39]). Actin-binding spectrins (239 and 100 kDa) are associated with the cell membrane, the nuclear envelope, the growing mitochondrial cloud (MC) and nucleoli [40,41], as well as intermediate filaments [42,43]. Isoforms of protein 4.1 (4.1R) stabilize the spectrin–actin network and anchor it to the plasma membrane [44]. Therefore, CSK is highly present in the process of the more complex organization in the early-stage oocyte. Cytoskeletal proteins may have a specific functional role during ribosome maturation and translation, being active in these stages of oogenesis [45,46]. Ribosomal subunits are produced in the nucleolus and then exported to the cytoplasm, where they associate with a vast number of essential factors, as well as with cytoskeletal proteins (see Chierchia et al. [47]): this process is very relevant in the *X. laevis* oocyte, which contains more than 1000 nucleoli [36]. Some proteins, including rpl10, the eukaryotic initiation factor 6 (eif6), thesaurins A/B and P0, are involved in 60S ribosomal maturation and/or translation [47]. eif6, specifically, is associated with pre-60S and is responsible for the regulation of 60S availability for translation and the formation of the active 80S ribosome [48,49]. During the maturation, specific coordinates are packed within the oocyte for the organization of animal/vegetal (A/V) polarity that later constitute the antero/posterior (A/P) axis of the embryo. For the acquisition of the A/V polarity, the mitochondrial cloud (MC) must be mentioned among the most important factors. At stage 1, the MC grows while anchored to the nuclear envelope, representing a clear sign of asymmetry in the oocyte; therefore, the virtual animal/vegetal axis of the oocyte bisects the nucleus and MC [50]. It can be defined as a package of RNA held together by a network of cytoskeletal proteins, endoplasmic reticulum and mitochondria [51], useful for the transport of genetic information to the oocyte cortex in a region facing the plasma membrane called the messenger transport organizer (METRO). MC is surrounded and sustained by a conspicuous network of cytoskeleton components (cytokeratin, spectrin, γ-tubulin and p4.1 [18,41,52]). The MC gradually migrates toward the vegetal cortex and releases its content into the proximal cortex, thereby marking it as the vegetal pole [53]. At the same time, an early event of cytoskeleton polarization occurs with an unknown mechanism. Actin and the α- and β-spectrins (mRNA and proteins) form a cap [54]. α-spectrin colocalizes with the mRNA of XNOA 36 (homologous to mammalian NOA 36 RNA) [55] and gradually segregates parallelly to the A/V axis, thereby virtually cutting the MC and the oocyte into two halves [41], showing the existence of a transient cytoskeletal-dependent process of mRNA localization/anchoring in stage I oocytes. In the “late pathway” of stages II–III of oogenesis, Vg1 and VegT mRNAs responsible for endoderm and mesoderm specification to the region marked by the MC content are transported to the METRO vegetal territory [37,53,56]. Here, the MTs gradually reorganize in the vegetal hemisphere of the oocyte and spread throughout the whole cytoplasm. When the animal and the vegetal splits of oocytes are formed, the oocytes contain an extensive network of principal cytoskeletal proteins with a radial organization, and the distribution of cortical γ-tubulin temporally coincides with the formation of the A/V axis12. In previous studies, the use of CSK-disruptive drugs suggested that the organization of the oocyte CSK depended upon a hierarchy of interactions among F-actin, MTs and CK filaments via linker proteins [42]. More recently, it was proposed that the 56 kDA of p4.1R may constitute a linker between the actin cytoskeleton and cytokeratin [44] and that there is a specific regionalized distribution of spectrin, as it accumulates in the cortex of the oocyte animal half and may anchor plasma membrane molecules involved in sperm binding and fusion [57].

### 4.2. Xenopus as a Model for the Study of Organogenesis

Since 1950, *X. laevis* has been pioneer animal model for the study of several organs and regeneration during embryogenesis (Figure 3). It has been advantageous for the study of the spinal cord, both during development and in the adult, for several reasons. The first is the simple organization of this organ, in contrast to that of higher vertebrates. In relation to this, it should be mentioned that *Xenopus* organs and cell cultures are ideal for long periods of live imaging because they are easily obtained and maintained, and they do not require special culture conditions. The neural tube–spinal cord organogenesis was studied, together with its cell specification and differentiation, spinal neuron axon guidance, neuromuscular junction formation and plasticity, as well as spinal cord injury and regeneration [58,59]. The mechanical forces were depicted that promote changes in cell shape and lead to the formation of the neural tube thanks to the combination of cellular and molecular mechanisms with biochemical signals [60,61]. Moreover, *X. laevis* at the tadpole stages exhibits a remarkable regenerative capacity: many tissues and organs, including the spinal cord, lens, tail and limbs, can be regenerated. This ability, however, decreases throughout metamorphosis progression and is almost completely lost after metamorphosis and therefore in adult life [62,63]. *X. laevis* has been also crucial in the identification of the mechanisms governing axon guidance. Localized variations in the expression levels of cyclic nucleotides were found to transduce the signals of guidance cues, able to change the direction of the growing axon [63,64,65]. *Xenopus*, if compared to other vertebrates, has rather large growth cones and it therefore constitutes a suitable system for the imaging of cytoskeletal components during neural development, when neurons elongate their axons to form new connections. The growing cones are exposed to morphogen gradients that activate signaling cascades, generating modifications in actin filaments (F-actin) and microtubule (MT) dynamics. As a result, the growth cone varies in morphology, controlling the axon movement [66,67]. Several studies contributed to the understanding of the functions of multiple cytoskeletal components, including actin, microtubules and neurofilaments, during axon pathfinding [68].

### 4.3. Interactions of Specific Proteins in Organ Formation: The Eye and Kidney

As already stressed, *Xenopus* is a strategic model for the study of early embryonic events, such as the determination and differentiation of organs such as the eyes and kidneys. This stems from the fact that in *Xenopus* RNA, the antisense localization of early factors determining the destiny of the embryonic field is easily performed and that the embryos rapidly recover from surgical manipulation. In the case of the eye, eye field specification takes approximately 14 h, and the final eye is formed in approximately three days. The specification of the eye field in the neural plate is regulated by several interacting signals. The use of antisense RNA and antisense morpholino oligonucleotide and other molecular manipulations significantly decreases the levels of gene products instrumental for eye formation, thus allowing us to understand their function in this process. Transcriptional factors expressed in the early eye field are specifically required for eye morphogenesis (EFTFs, such as Pax6, Six3, Rx1, etc.) [69,70]. They are expressed together in *Xenopus*’ anterior plate before stage 15 [71], and their genetic manipulation constitutes an important tool in determining their complex interplay for eye determination and differentiation. From the eye field will originate the optical cup and then the optical vesicle with a mechanism quite similar to that in other vertebrates. Which are the early signals determining eye formation? Two studies [72,73] demonstrated that the insulin growth factor (IGF) is sufficient for the formation of the anteriormost structures, including the eyes. This because the IGF pathway regulates head formation by antagonizing Wnt signal transduction at the level of β-catenin in early *Xenopus* embryo development [74]. During early embryogenesis, the signal derived from IGF is maintained by GAIP interacting protein, C terminus (gipc2) [75]. In particular, eif6 regulation of gipc2 enables the correct morphogenesis of the *Xenopus* eye [7]. Is there a relationship between the two molecules? In fact, eif6, such as gipc2, is part of the pi3k/akt branch of IGF signaling, since the overexpression of eif6 downregulates the activation of akt. For this reason, the eif6 eye phenotype is related to the impairment of the IGF pathway because of the interaction of eif6 with IGFr along the same pathway where gipc2 is active. Moreover, the action of eif6 on gipc2 is direct because eif6 mRNA downregulates gipc2 expression in a dose-dependent manner, indicating that eif6 negatively regulates the expression of gipc2 mRNA. Then, eif6 may control the expression of gipc2, which is essential for eye morphogenesis, suggesting that the role played by eif6/gipc2 partnership for the morphogenesis of embryo organs is dependent onIGF-induced akt activation [7]. Studies on the genetic and morphological patterns of renal development in *Xenopus* showed us that, similarly to the eyes, the process of kidney development is conserved among vertebrates. Development of the kidney occurs through a series of three increasingly complex nephric structures that are temporally distinct and occupy discrete spatial sites within the body. These systems serve to provide homeostatic, osmoregulatory and excretory functions in animals. Importantly, in amphibians such as *Xenopus*, the kidney (called pronephros) is less complex and more easily accessible than the metanephric kidney in mammals. Thus, tadpoles and frogs provide useful models for the study of kidney development [76]. In particular, in *X. laevis*, several signaling cascades contributing to pronephric kidney development appear to be well conserved during mammalian metanephric development [76]. lim1 (lhx1), a member of the LIM family of homeobox genes, is one of the earliest known genetic markers of kidney development [77]. In *X. laevis*, the expression of this transcription factor is critical for the specification of cells to a renal fate [78,79] and for the promotion of tubule growth and elongation as development progresses [80]. In the murine system, the same phenomenon is observed: lim1 is in fact essential for multiple steps in the morphogenesis of epithelial tubules in metanephric kidney development [81,82]. The ability of eif6 to control gipc2 expression is active also in the case of pronephros development, where eif6 overexpression causes edema and damage of the three main components of the pronephros. gipc2 depletion induces the edema phenotype and reduction of the pronephros constituents, as indicated by several markers, such as lhx 1 and nephrin [8]. Moreover, p110*, a constitutively active p110 subunit of pi3 kinase, corrects the abnormal kidney phenotype induced by eif6, suggesting that, similarly to the eye, eif6 may control the expression of gipc2, which is essential for pronephros morphogenesis and dependent on igf-induced akt activation [8].

### 4.4. Xenopus and Studies of Apoptosis in Development

During *Xenopus* embryogenesis, the construction of the body is conducted through proliferation and programmed cell death (PCD). Apoptosis occurs at the beginning of embryogenesis and in organogenesis [83,84,85] and is required for the removal of the tail in anurans during metamorphosis and of the interdigital membrane in the developing vertebrate limb [86]. During *Xenopus* embryogenesis, apoptosis is activated at the beginning of gastrulation and follows a spatiotemporal pattern, which has been thoroughly described until the tadpole stage [87]. In early development, apoptosis is identified only within the presumptive neuroectoderm at the neural plate stage. Later, apoptosis remains localized to specific brain regions, giving rise to sensory organs and to the spinal cord. Moreover, numerous neurons die via apoptosis, in the process of building proper synaptic organization and axonal pathways [87]. Cell death and cell survival are regulated by complex and interacting signals. Data from Finkielstein et al. [88] and Yeo and Gautier [89] indicated that after middle blastula transition, the Bcl-2 and Bax balance and downstream caspase activation are the main mechanisms involved in the control of apoptosis. Moreover, the spatiotemporal pattern of the expression of several genes regulating early neural development coincides with that of apoptosis, suggesting a role for apoptosis in primary neurogenesis [87]. Apoptosis is required for normal neural development and some neural cells display apoptosis as part of their normal neural differentiation [89]. Which factors control the occurrence of apoptosis during normal development? In early *Xenopus* embryogenesis, eif6 is among the proteins involved in the maintenance of apoptosis levels [90,91]. eif6 is expressed in variable amounts in developing anlagen, in addition to a basal level of expression. In particular, eif6 mRNA is copious in areas with high proliferation levels, such as the eye and mid-hindbrain periphery. eIF6 mRNA and protein levels are in fact detected in rapidly proliferating cancer cells and in low levels in cells committed to apoptosis; it seems to be part of an essential mechanism for the translation and reversal of cell death to cell survival, thus regulating the apoptotic process during normal development [91].

## 5. *Xenopus* in Environmental Studies

### 5.1. Xenopus in the Study of Microplastic Environmental Pollution

In recent years, *X. laevis* has been used to study the effects of microplastics (MPs), emerging environmental contaminants that display the ability to reach organs vital to organisms, causing several impairments. Although the main sources of MPs in marine environments are inland surface waters, information on the occurrence and the effects of MPs in freshwater ecosystems is still scant. Pekmezekmek et al. [92] recently demonstrated that adult *X. laevis*, if fed with polyvinyl chloride (1% of body weight, twice each week for 6 weeks) before the fertilization period, gave birth to embryos with evident malformations and decreased viability. In addition, the Hsp70 and Pax6 gene expression levels significantly decreased, although the lung and intestine tissues displayed a normal appearance in the histopathological examination. This is because, if ingested with a natural food source, microplastics act as artificial fibers that reduce food quality by decreasing nutrients and energy, with possible impacts on growth and development. Ruthsatz and coworkers [93,94] showed, in the same species, that the embryonal gut length and mass increased in response to microplastic ingestion, indicating that MPs induced digestive plasticity—it is likely that they fully compensated for the low nutrient and energy density by developing longer intestines. In 2023, Ruthsatz et al. [95] showed that MP ingestion resulted in sublethal effects on embryo growth, development and metabolism and led to allometric carryover effects on juvenile morphology, accumulated in the specimens at all life stages. Embryos exposed from stage 36 to 46 to 0.125, 1.25 and 12.5 μg mL^−1^ of polystyrene MPs (PSMPs) showed the presence of MPs in their digestive tracts but not in the gills, for each tested concentration. Moreover, neither body growth nor swimming activity was affected by PSMP exposure. These data indicate that PSMP ingestion does not alter *X. laevis* development and swimming behavior during early life stages [96]. To date, most studies of microplastics (MPs) have focused on single commercial micro- and nano-polymers, thus not reflecting the heterogeneity of environmental MPs. To overcome this limitation and to improve the environmental hazard assessment, miscellaneous waste plastics of a micrometer size (<100 µm and <50 µm) were characterized and tested in developing *X. laevis* and zebrafish, respectively, by the FETAX and FET assays [97]. In zebrafish, miscellaneous MPs caused a low rate of mortality and altered phenotypes in embryos, establishing species-specific bio-interactions. In *Xenopus* embryos, MPs were accumulated in the intestinal tract, producing mechanical stress and mucus overproduction, but not interfering with morphogenesis processes. Bacchetta et al. [98] analyzed the effect on polyester fibers taken from a dryer machine. They showed that the gastrointestinal tracts of *X. laevis* embryos were the most affected systems, with physical occlusion observed in a concentration-dependent manner. However, no other damages were registered. No mortality was observed, but exposed embryos showed a significant reduction in their mobility. Generally, as bottom, non-selective grazers, the free-living young life stages of amphibians may be particularly vulnerable to sedimented plastics. In the literature, a few studies have assessed amphibians’ exposure to MPs through contaminated food or their effects in feeding behavior assays. Venancio et al., in 2022 [99], described major differences found in the available studies, aiming to highlight the importance of feeding exposure assays when attempting to evaluate the effects of MPs in amphibians.

### 5.2. Xenopus in the Study of Embryo Toxicology

Evidence has shown that *Xenopus* embryos are sensitive to environmental compounds [13,100]. In ecotoxicology and reproductive toxicology laboratories, the FETAX assay is routinely employed to investigate mortality, malformations and modifications in molecular mechanisms in response to various environmental factors [101,102,103,104,105] (Figure 1). Using this assay, several chemical environmental pollutants (such as benzene, methylmercury, cadmium, zinc and polychlorinated biphenyls) have been found to cause neurodevelopmental toxicity in *Xenopus* embryos [106,107]. Para-xylene (PX), through the methylation and acetylation of histones, can cause neural apoptosis and developmental delays in the developing *Xenopus* [108], also inducing deficits in neuronal structure [109]. Ethylbenzene inhibits NMDA-induced currents [110] and acetylcholine-induced ionic currents [111] in *Xenopus* oocytes expressing NMDAR or a9a10 nAChR. Several types of pesticides, including herbicides, insecticides, fungicides and rodenticides, efficient in pest control and crop productivity, were proven to be highly toxic, with serious impacts and consequences for human health as well as aquatic ecosystems [112]. Several distinct chemical classes of pesticides have been primarily investigated in *Xenopus*, such as chlorpyrifos (organophosphorus insecticide), thiacloprid (insecticide), atrazine (triazine herbicide) and triadimefon (conazole fungicide) [113,114,115,116,117,118]. All these pesticides were proven to exert differential toxic effects on embryogenesis, neurotransmitter release and sexual differentiation. In particular, pyrethroid insecticides activate dopaminergic, noradrenergic and serotonergic neurotransmitter systems, leading to interference in metamorphic development [119], and, among the organophosphorus insecticides, dichlorvos was found to decrease the heart rate, spine length and free-swimming activity of *Xenopus*, and chlorpyrifos produced a dose-related increase in tadpole teratogenic alterations, resulting in a reduced growth rate, locomotor activity, heart rate and survival ability and altered melanocyte distribution [114,120]. The dispersion of skin pigment is a marker of chemical toxicity in water ecosystems. The first study to assess chemical toxicity in drinking water was performed in cultured *Xenopus* melanophores [121]. They found that some chemicals, including ammonia, arsenic, copper, mercury, pentachlorophenol, phenol, nicotine and paraquat, induce a rapid melanosome dispersion response, indicating that *Xenopus* melanophores may serve as a good toxicity sensor in screening toxicants. Pharmaceutical residues also pose a serious threat to environmental health. In the past three decades, pharmaceutical residues have been in fact discovered in almost all environmental matrices on every continent [122,123,124]. Their residues in the environment are considered to be “compounds of emerging concern” and they are becoming ubiquitous in the environment because they cannot be effectively removed by conventional wastewater treatment plants [125]. Several widely used pharmaceuticals are therefore routinely found in sewage treatment plant influents and effluents and ultimately in surface water, thus reaching a plethora of non-target organisms. Considering its reliability and reproducibility, FETAX was validated as a useful tool in predicting the toxicity and teratogenicity of pharmaceuticals [126]. To date, many pharmaceuticals commonly found in the environment have been investigated in *Xenopus*, from selective serotonin reuptake inhibitors (SSRIs) [127,128] to anticancer [129] and non-steroidal anti-inflammatory drugs (NSAIDs) [130,131] and psychotropic drugs [132], revealing strong teratogenic power and robust interference with many crucial aspects of embryonic survival. *Xenopus* represents an efficient model for the in vivo study of environmental toxicity in embryo development; for this reason, the use of this model rather than the conventional ones has increased over the years (Figure 4).

## 6. Xenopus in Emerging Areas of Bioscience

### Nanoscience and Xenopus Developmental Studies

With the advent of the nanotechnological era, many biological fields have been adapted with the goal of discovering the benefits and hazards of new nanomaterials (NMs) in several biosystems. According to the 3Rs principle, in vitro systems have been mainly promoted by health authorities to sustain this research, but continuously increasing evidence points out that in vivo experimentation is of extreme relevance in nano-bioscience, mainly due to the still unpredictable behavior and biological mechanisms of action of the new NMs. In this scenario, extensive efforts are today devoted to the development of complex 3D in vitro models, although the use of alternative animal models should also be considered a priority for the prediction of NM toxicokinetics and toxicodynamics. However, as underlined by Hussein and colleagues [133], in vitro nanotoxicology is currently at a crossroads and many influential challenges need to be addressed. The following have been listed: (1) more rigorous NM characterizations; (2) the correlation of NM physicochemical parameters and toxicity; (3) dose–response behavior under relevant environmental and human exposure; (4) the selection of cellular models relevant to exposure and predictive of the in vivo response; (5) assessments covering acute and chronic exposure. One of the possible strategies to overcome these gaps is through the development of advanced in vitro models, considered more representative of the NM interactions and effects in vivo. Especially for inhalation and ingestion toxicology and bio-interaction purposes, huge efforts are being made to standardize the use of 3D reconstructed tissues, incorporating immune cells and including the use of physiologically relevant fluids [134,135]. In parallel with the enhancement of the in vitro systems, the use of non-mammal vertebrate organisms has been taken into consideration as valid alternative models for NM screening [136]. In particular, in the last few years, many groups around the world have achieved success in using zebrafish as an in vivo alternative model for both nano-EHS and nanomedicine studies [137,138]. This model shows many advantageous features that have contributed to its success. Moreover, among the alternative vertebrate models, *X. laevis* should also be included, given its historical use in many cell and developmental biology labs, as well as its relevance in environmental toxicology studies. Nevertheless, relatively little consideration has been given to this model in the recently advancing sectors of nanotoxicology and nanomedicine.

Since 2001, *Xenopus* has been considered a good model regarding the effects arising from the use of nanomaterials (Figure 5). It is now known that nanoparticles can produce toxicity and genotoxicity by interfering with gene expression in various ways [101,102,139,140]. To date, only a few articles (circa 40 in 22 years) have been published dealing with the influence of nanomaterials on *X. laevis* embryo development. In this regard, we must note that *Xenopus* is a valid model to study developmental pathways and that it could be transferred to higher vertebrates and humans. In this scenario, Tussellino and coworkers’ paper, published in 2015, is relevant [101]. They showed that in *X. laevis*, following treatment with polystyrene nanoparticles, both with the FETAX assay and microinjection, the expression of some genes involved in early embryonic development was altered—in particular, the expression of bra (early mesoderm marker), myod1 (paraxial mesoderm marker) and sox9 (neural crest migration marker)—concluding that NPs modified the normal migration of the prospective axial and paraxial mesoderms and of the neural crest, namely the fourth germ layer. The same authors [102] have shown that, in the same species, silica nanoparticles also modify gene expression in early embryos. Carew et al. [140], using a MAGEX microarray and quantitative real-time PCR, showed, in *Xenopus* tadpoles, that AgNPs induced the disruption of five TH-responsive targets and an increase in the mRNA of two peroxidase genes. These NPs had developmental-stage-specific endocrine TH-dependent disrupting effects during metamorphosis. Moreover, *X. laevis* is an excellent biological model for bioimaging to study morphological and physiological dynamics, both of cells and organisms, thanks to its particular characteristics. In this context, few papers utilizing mainly QDs have been published. Dubertret et al., in 2002 [141], produced fluorescent QDs suitable for lineage-tracing experiments in embryogenesis. To address the problem of biocompatibility, they encapsulated individual nanocrystals in phospholipid micelles and demonstrated that these QDs conjugated to DNA were stable and nontoxic, both in vitro and in vivo, and thus suitable for imaging. In 2009, Stylianou and Skourides [142] showed, both in vitro and in vivo, that focal adhesion kinase inhibition, without affecting convergent extension, blocks mesoderm spreading and migration. They utilized near-infrared QDs. Quantum dot nanocrystals and polystyrene beads were injected into embryos of *X. laevis* to study the development of larval brain ventricles and the patterns of cerebrospinal flow [143]. Brandt et al. [144] utilized the QDs’ fluorescence capabilities for the real-time visualization of Cyclin E trafficking within *X. laevis* cells. They showed that the bonding of QDs to Cyclin E did not affect its intracellular distribution during the mid-blastula transition. In 2019, Umanzor-Alvarez et al. [145] set the parameters for near-infrared laser injection able to provide QDs, with the same effectiveness as manual injections, to *Xenopus* embryos without damaging their vitality and morphology. Galdiero and coworkers [146] showed that *Xenopus* is also a useful in vivo model for toxicity and drug delivery studies. Pigments utilized in tattooing, such as RP170, having nanoparticle dimensions, produced in *X. laevis* embryos modifications of survival and a higher malformation rate and ROS production, probably acting on gene expression [147].

The perspectives opened up by the studies described above are very broad and it would be highly useful to consider them for future studies aimed not only at the knowledge of the *Xenopus* system but at the acquisition more generally of the physiological and morphological mechanisms that take place in an organism, using *Xenopus* as a model.

## 7. Xenopus in Immunology, Genetic and Disease Studies

### 7.1. Xenopus laevis as a Model for Human Genetic Disease

Novel advances in scientific methodologies have allowed the investigation of genetic diseases in organ-specific contexts, but also in several syndromes [148,149,150,151,152]. Also in this field, *X.laevis* over the years, has made his contribution to research (Figure 6). In general, to generate and analyze a certain condition, the antisense-based morpholino (MO) knockdown approach is the most used. In *X. laevis* embryos, thanks to their large size and the relatively long time from their fertilization to first division [153], MOs have usually been injected in order to study the progress of development. In 2022, Treimer et al. [154] used morpholino tp53rk knockdown, showing that the loss of endogenous Tp53rk caused abnormal eye and head development. TP53RK encodes for one component of the KEOPS complex, defects of which represent the major cause of GAMOS, or Galloway–Mowat syndrome (GAMOS; MIM# 251300). Moreover, Mann et al. [155], in 2021, found a mutation in PRDM15 that was a novel cause of Galloway–Mowat syndrome. This finding was possible thanks to the morpholino-oligonucleotide-mediated knockdown of Prdm15 in *X. laevis* embryos, which hindered pronephric development. MO antisense oligomers were used also for congenital primary hypothyroidism (CH), a common endocrine disorder in neonates. The researchers identified the most promising candidate gene, transient receptor potential channel 4-associated protein (TRPC4AP), and they studied its function in *X. laevis* [156]. In recent years, CRISPR/Cas9 knockouts (KO) have been increasingly used to study gene function. In particular, CRISPR/Cas9 technology is available for the screening of candidate disease genes and creation of patient-specific mutations for human disease in *Xenopus* [28,157,158]. An advantage of working with *X. laevis* is the large body of literature on the genotype/phenotype correlations and the availability of injection and tested protocols optimized by the large community of researchers using *X*. *laevis* frogs as a model system in development and many other fields of research. Oculocutaneous albinism in humans (human synonym OCA1) was obtained via the knockout of the *Xenopus* tyrosinase gene, producing similar phenotypic outcomes, but it requires the co-targeting of two gene homeologs because of the allotetraploidy of *X. laevis* [159]. Viet et al. [160], in 2020, characterized the developmental process of lens formation in *Xenopus laevis* tailbuds and tadpoles by the disruption of orthologs of three mammalian-cataract-linked genes in F0 by CRISPR/Cas9. *X. laevis* was used also to better understand the encephalopathies. In fact, Sega et al., in 2018, found a novel mutation in neuronal differentiation factor 2 (NEUROD2). To model the disease and assess the functional effects of patient variants on candidate protein function, researchers used in vivo CRISPR/Cas9-mediated genome editing and protein overexpression in frog tadpoles. Depleting neurod2 with CRISPR/Cas9-mediated genome editing induced spontaneous seizures in tadpoles, mimicking the patients’ condition [161]. The principal aim of this technique is determining the biological relevance of all variants, as many candidate genes have no known role in disease. For this reason, animal models, particularly *X. laevis*, offer the opportunity to discover the molecular function of each variant.

### 7.2. Xenopus laevis in Immunologic Studies

Since 1964, a large number (1264, Figure 7) of papers have been published using *X. laevis* both to study the immune system of this animal and, in recent years, as a model for immunology studies (210 papers in the last 10 years and 107 in the last 5 years). Therefore, we will limit ourselves to considering the papers that we found most interesting in the last 10, considering the various aspects of the immune system of *X. laevis*. Comparative, nonmammalian animal models are important both to gain a basic evolutionary understanding of the complex interactions of the immune system and to better predict new immunotherapeutic approaches in humans. *X. laevis* is a valuable alternative model in the study of the immune system. T cell development occurs in the thymus, which is a primary lymphoid tissue [162,163]. In amphibians, thymectomy is easy [164], and *X. laevis* is a useful model for the study of larval T cell persistence post-metamorphosis [165] and T-cell-dependent adaptive immunity in a vertebrate tetrapod [166,167,168]. The migration of T cells from the thymus, particularly γδ T cells, occurs relatively early in development, as shown by Foulkrod et al. [169], who found differential expression of the TCR gene, expressed in γδ T cells, in different organs, but also differential expression between the larval and adult stages. The authors concluded that since *X. laevis* is placed in an intermediate position between fish on the one hand and chickens and mammals on the other, the data obtained on this amphibian provide a necessary basis for future studies of γδ T cell functions [169]. McGuire et al. [170] showed the effects of a mixture of organic compounds composed of naphthalene, ethylene glycol, nonylphenol ethoxylate and octylphenol on the production of thyroid hormone (TH), the destruction of which induces problems in thymocyte production. Exposure to this mixture not only induced thyroid hypertrophy and delayed metamorphosis but also reduced the number of immature cortical specific thymocytes (CTX+) and mature CD8+ thymocytes in *Xenopus*. This situation was recovered if animals were simultaneously exposed to exogenous TH (T3). Mycobacterial infections represent major concerns for aquatic and terrestrial vertebrates, including humans. In mammals, protective immune responses involving the innate and adaptive immune systems are highly complex and therefore not fully understood. As with a mammalian fetus or infant, T cells and myeloid cells in *X. laevis* tadpoles differ from adult immune cells. Larval or fetal T cells originate from progenitors distinct from those of adult immune cells. Some early developing T cells and subpopulations of macrophages are conserved in anurans and adult mammals, probably playing an important role in host–pathogen interactions from early development to adulthood [171]. Invariant T cells (iT) have gained attention recently because of their potential as regulators of immune function. These cells undergo differentiation pathways distinct from those of conventional T cells. *X. laevis* was used to characterize a population of innate T cells (iT) by transgenic techniques, which allowed the identification of a subset of iT, CD8(−)/CD4(−), in both tadpoles and adults. This population is critical for antiviral immunity during the early larval stages. The authors concluded that mucosa-associated invariant T cells in *X. laevis* show the importance of iT cells in the emergence and evolution of the adaptive immune system [172]. *Mycobacterium marinum* is a pathogen that infects many vertebrates, including humans, and is problematic for aquaculture and public health. Rhoo and coworkers [173] developed a model of infection in *X. laevis* to study host responses to *M. marinum* infection in two distinct life stages, tadpole and adult. It was hypothesized that adult frogs possess efficient conventional T-cell-mediated immunity, whereas tadpoles rely predominantly on iT cells. *Chytridiomycosis* (*Batrachochytrium dendrobatidis*) is considered an emerging infectious disease, linked to the decline and extinction of amphibians worldwide. Although amphibians have well-developed immune defenses, clearance of this pathogen from the skin is often impaired by factors (methylthioadenosine, tryptophan and an oxidized tryptophan product, kynurenine) that inhibit the immune system by inhibiting the survival and proliferation of lymphocytes in amphibians and human Jurkat T cells. It is likely that *B. dendrobatidis* adapts its metabolism to release products that alter the skin environment to inhibit immunity by enhancing its own survival [174]. Hyoe et al. [175] studied the role of iT cells during mycobacterial infection by developing reverse genetics and MHC tetramer technology to characterize an MHC/iT-like system in tadpoles. The study provided evidence of a conserved convergent function of iT cells in host defenses against mycobacteria in amphibians and mammals. *X. laevis* was also used as a model to study the role of iT cells and the interaction of class I major histocompatibility complex (MHC) molecules in tumor immunity. In *X. laevis*, two specific subsets of iT cells, iT cells Vα6 and Vα22, recognize and fight tumor cells. In addition, the complex functions of the XNC10 gene of *X. laevis* in tumor immune responses have been shown. Banach and Robert [176], in a recent study, aimed to better understand the activity of natural killer T (NKT)-cells against mammalian tumors, and they studied iT cells of *X. laevis* and the interaction of MHC class I genes (XNC or mhc1b.L) against ff-2 thymic lymphoid tumors. They demonstrated the relevance of the XNC10/iT cell axis in controlling tumor tolerance or rejection. In *X. laevis*, the differentiation of distinct leukocyte subsets is governed by specific growth factors that elicit the diverse expression of transcription factors and markers by developing cell populations. For example, macrophages (Mφs) and granulocytes (Grns) are derived from common granulocyte–macrophage progenitors in response to distinct myeloid growth factors. In turn, myelopoiesis appears to be similar to that in other vertebrates, while the functional differentiation of Mφs and Grns from their progenitor cells remains poor, although there appears to be an evolutionary kinship among the various amphibian myeloid lineages [177]. Myeloid progenitors reside within specific hematopoietic organs. Unlike other vertebrates, in *X. laevis*, macrophage precursors are absent from the liver and instead reside within their bone marrow. The data indicate that *X. laevis* retains its megakaryocyte/erythrocyte potential in the hematopoietic liver while sequestering its granulocyte/macrophage potential in the bone marrow, thus marking a new myelopoietic strategy compared to those of other vertebrate species [178]. Adipose tissue macrophages (ATMs) have a metabolic impact in mammals because they contribute to metabolically damaging adipose tissue inflammation. Controlling the number of ATMs may have therapeutic potential; however, information on the ontogeny of ATMs is scarce. While ATMs are thought to develop from circulating monocytes, various tissue-resident Mϕ are capable of self-renewal and develop from progenitors independent of a monocyte intermediate. Waqas et al. [179] showed that the adipose tissue of *X. laevis* contains self-renewing ATMs before the onset of hematopoiesis. The data obtained indicate that ATMs have a prenatal origin and that such a study in *X. laevis* may aid in understanding the environment in which ATMs develop. In the past decade, it has become evident that the innate immune system is able to influence the speed and quality of the regenerative response by unclear mechanisms—hence the interest in studying the role of inflammation during tissue repair and regeneration. *X. laevis* embryos, which are able to achieve scarless wound healing and regenerate appendages, are useful in understanding the role that inflammation plays during tissue repair and regeneration in this organism. Characterization of the morphology and migratory behavior of granulocytes and macrophages was performed using various transgenic lines that served to mark myeloid lines, including granulocytes and macrophages. The data showed that the inflammatory response following injury and infection in *X. laevis* larvae is very similar to that observed in humans [180]. Infections with ranaviruses (Frog 3; Fv3) are contributing significantly to the decline of the global amphibian population. In particular, amphibian macrophages (Mφs) are important for both Fv3 infection strategies and immune defense against this pathogen. However, the mechanisms remain unknown. In addition, other factors, such as interleukin-34 (IL-34) and CSF-1, are involved in immune defense [181]. While tadpoles of anuran amphibians were previously believed to be more susceptible to Fv3, Kalia et al. [182] showed that tadpoles are actually more resistant to this virus than metamorphic and postmetamorphic frogs. This resistance is conferred by large myeloid cells within the tadpole kidneys, which show the high expression of endogenous retroviruses. In turn, these endogenous retroviruses activate cellular double-stranded RNA-sensing pathways, resulting in increased antiviral cytokines and interferon expression, thus providing the observed anti-Fv3 protection. XLC-1 lectin from *X. laevis* of the intelectin family selectively recognizes carbohydrate chains on the bacterial cell surface. The intraperitoneal injection of lipopolysaccharide (LPS) from *Escherichia coli* caused an increase in XCL-1+ cells in the spleen at day 3, followed by a rapid decrease by day 7. The same response was observed in peripheral blood leukocytes and peritoneal exudate cells. XCL-1+ cells showed the morphological characteristics of macrophages. The results indicate that in *X. laevis*, macrophage-like cells produce XCL-1 and XCL-1 facilitates the phagocytosis of bacteria [183]. The glutamic acid–leucine–arginine (ELR) sequence is a distinctive feature shared by mammalian inflammatory CXC chemokines such as interleukin-8 (IL-8; CXCL8). *X. laevis* encodes multiple isoforms of CXCL8, one of which (CXCL8a) possesses an ELR motif, while another (CXCL8b) does not. These CXCL8 isoforms show distinct expression patterns during frog development. Koubourli et al. [183], using recombinant forms, defined their functional differences.

## 8. Conclusions

In this review, we summarized the main strengths of *X. laevis*, a versatile model to address many biological and environmental issues. Hence, we strongly recommend improving its use in the basic and applied research devoted to the safety assessment of the environment and health.

## Figures and Tables

**Figure 1 biology-12-00890-f001:**
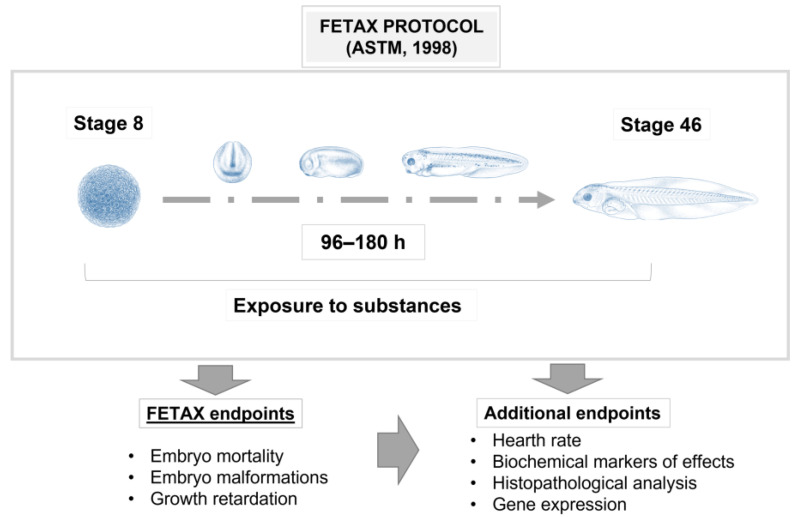
Scheme of Frog Embryo Teratogenesis Assay—*Xenopus* (FETAX), employed to measure the activity of pollutants during embryo development. *Xenopus* illustrations © Natalya Zahn (2022) and sourced from Xenbase (www.xenbase.org RRID: SCR_0032803; Accessed on 12 January 2023) [15].

**Figure 2 biology-12-00890-f002:**
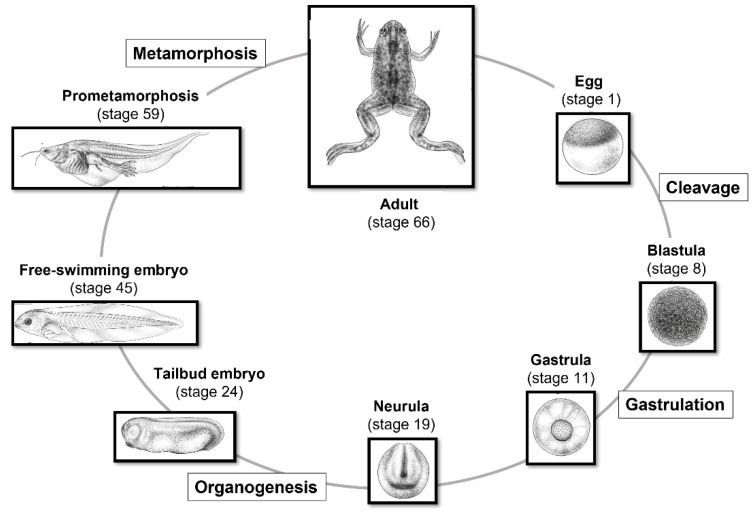
Life cycle of *X. laevis* staged according to Nieuwkoop and Faber (1956) [6]. Through in vitro and in vivo fertilization, it is possible to follow all stages of early embryonic development. *Xenopus* illustrations © Natalya Zahn (2022) and sourced from Xenbase (www.xenbase.org RRID: SCR_0032803; Accessed on 12 January 2023) [15].

**Figure 3 biology-12-00890-f003:**
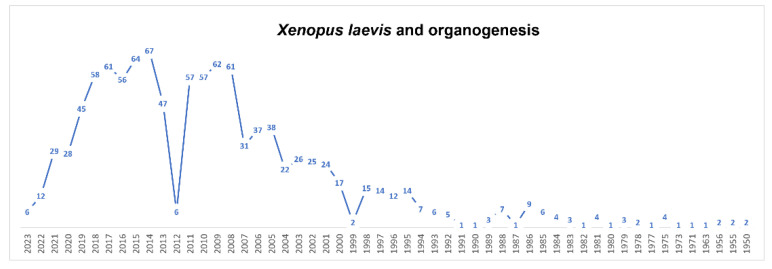
Timeline of published papers (number per year) from the PubMed website using the keywords “*Xenopus laevis*” and “organogenesis” (updated to March 2023).

**Figure 4 biology-12-00890-f004:**
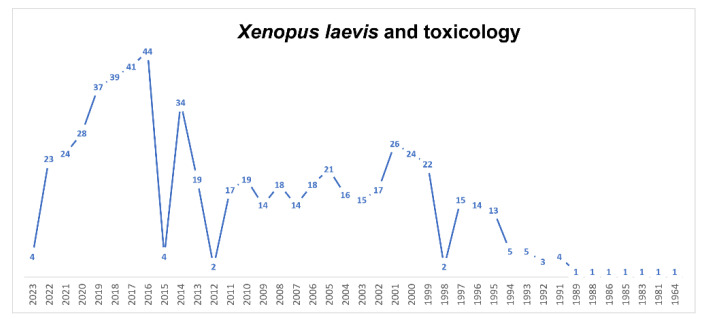
Timeline of published papers (number per year) from the PubMed website using the keywords “*Xenopus laevis*” and “toxicology” (updated to March 2023).

**Figure 5 biology-12-00890-f005:**
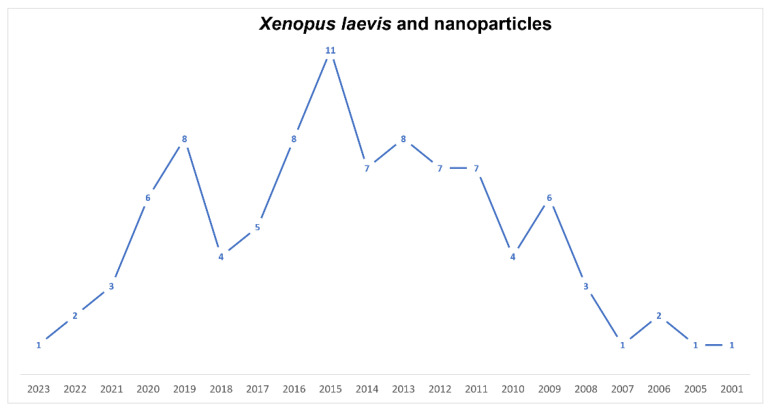
Timeline of published papers (number per year) from the PubMed website using the keywords “*Xenopus laevis*” and “nanoparticles” (updated to March 2023).

**Figure 6 biology-12-00890-f006:**
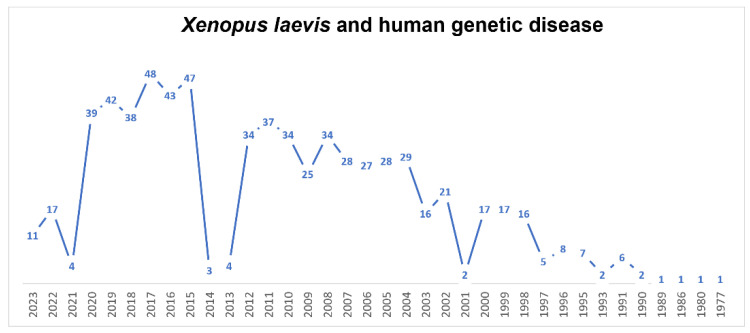
Timeline of published papers (number per year) from the PubMed website using the keywords “*Xenopus laevis*” and “human genetic disease” (updated to March 2023).

**Figure 7 biology-12-00890-f007:**
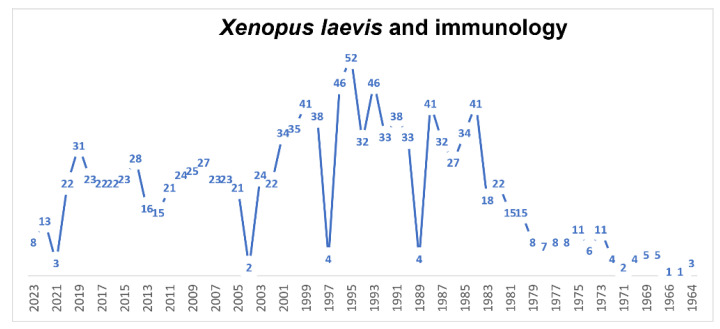
Timeline of published papers (number per year) from the PubMed website using the keywords “*Xenopus laevis*” and “immunology” (updated to March 2023).

## Data Availability

The data are contained within the article.

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
