# Peer review of "Xenopus laevis (Daudin, 1802) as a Model Organism for Bioscience: A Historic Review and Perspective"

_biology, 2023, doi:10.3390/biology12060890_

Round 1

Reviewer 1 Report

This is a review discussing the use of the amphibian Xenopus laevis as experimental organism for different areas of bioscience. The review includes sections on the use of X. laevis for nanotechnology, biological effects of microplastics and toxicology, which are areas not often discussed in other reviews. Thus, this current review has merit. 

However, there are several issues that the authors should consider.

1)  The text is unnecessary long due to useless repetitions and redundancies in the introduction and several following sections. The review would benefit to shorten these sections by deleting unnecessary repetitions.

2) The authors have the tendency to cite review rather than relevant primary literature. The authors should be careful to credit primary work for each important contributions of X. laevis in the area the discuss.

3) The authors use without clear rational and consistency Xenopus (the genus name), Xenopus laevis, X. laevis and X. tropicalis (species names). This is very confusing, especially since according to the title this is a review on Xenopus laevis. In addition, there is inconsistency in italicizing Xenopus and laevis or tropicalis. All species’ names should be italicized. After mentioning the Xenopus laevis fully, the abbreviation X. laevis should be used systematically in the whole text.  

4) Regarding the list of relevant contributions to scientific fields by X. laevis, the authors should consider adding immunology. Also, since this list is repeated several times in different sections, a table would be useful and prevent these repetitions.

5) The English would merit to be improved over the whole document. There are multiple inaccuracies, redundancies, and grammatical mistakes.

Other minor points

1) Simple summary, line 10-11 is a bit awkward. Suggest: Among advantages of this species one can mention:… 

2) Abstract line 19: it would be relevant to add immunology. 

3) Abstract, sentence line 21-22: delete “still”. Also, it is unclear what the authors mean by “diffuse”. Zebrafish is a useful and powerful alternative model, so it doesn’t seem appropriate or productive to be condescending or pejorative. Please revise.

4) Abstract, line 27: I don’t think that “higher genetic homology” is correct. Gene homology refers to an inference of common ancestor between two genes. This is not a metric. Suggesting changing by: “the X. laevis genome displays a higher degree of similarity with that of mammals.” 

5) Introduction, line 35: African clawed frog (add clawed), line 36 “has been” (not was), line 38 “many” eggs

6) Line 42-43: the authors may want to consider adding immunotoxicology. Several studies in X. laevis have been published in mainstream scientific journals over the last 10 years.

7) Line 50: delete the end of the sentence (and therefore...) that is redundant.

8) Line 82: Start a new sentence: “It was this ability…”

9) Figure legend 2 should use specifically X. laevis (all in italic) not Xenopus.

13) Line 113: Rana pipiens in italic

14) Line 118: X. laevis is fully resistant to Chytrid fungi (2 species), whereas X. tropicalis is susceptible. The authors should be accurate in their text. Other more relevant pathogens should also be mentioned including ranaviruses and mycobacteria (with references).

15) Line 126-127: this is a useless repetition from the introduction. Important references on tumor models (e.g., several well characterized transplantable lymphoid tumors) are missing, and again the extensive contribution of X. laevis to developmental immunology and immuntoxicology should be considered.

16) Line 136-136: Again, the use of homology in this sentence should be defined in more details. What 79% homology of genome means? Is this related to the number of gene homologs or the extent of sequence identity?

17) End of section 3. Most reference cited about cancer studies in X. laevis are reviews. The authors should cite primary references not just reviews. Besides studies in embryos, lymphoid tumors and anti-tumor responses have been extensiuvely characterized in X. laevis. This work cannot be omitted.

18) Section 5, line 228, one can perhaps also mention that tissue culture and live imaging for X. laevis can be done at room temperature (although optimal temperature for X. laevis cell lines is 27ºC), thus not necessitating special devices to warm up at 37ºC.

19) Sentence line 233- 234 regarding regeneration needs to be more specific and detailed. The regeneration capacity in X. laevis occurs from tailbud stage to pre-metamorphic stage 53 but is transiently lost from stage 45-47. Two critical recent work on this by J Gurdon group in Science 2019 and Development 2020, should be cited. 

20) Section 9, line 420 “some studies have been published. For example, Pekmezekmek… recently reported (not proved!). Also please split this sentence that is far too long and confusing.

21) Line 479-480: the authors should consider to also cite the drastic effects of atrazine and endocrine disruptor chemicals associated with hydrofraking on immunity in X. laevis

22) It should be noted that in first figure (Visual summary?) before the introduction the frog depicted is not X. laevis but X. tropicalis. Since this review is focused on X. laevis as shown in the title, it would be better to find a drawing or picture of X. laevis.

 English would merit to be improved over the whole document. There are multiple inaccuracies, redundancies, and grammatical mistakes. I have listed some mistakes and made some suggestions, but I think more editing work is needed. 

Author Response

This is a review discussing the use of the amphibian Xenopus laevis as experimental organism for different areas of bioscience. The review includes sections on the use of X. laevis for nanotechnology, biological effects of microplastics and toxicology, which are areas not often discussed in other reviews. Thus, this current review has merit. 

However, there are several issues that the authors should consider.

1)  The text is unnecessary long due to useless repetitions and redundancies in the introduction and several following sections. The review would benefit to shorten these sections by deleting unnecessary repetitions.

We did our best to delete repetitions.

2) The authors have the tendency to cite review rather than relevant primary literature. The authors should be careful to credit primary work for each important contributions of X. laevis in the area the discuss.

We have eliminated superfluous citations, preferring research articles, thus also reducing the number of total references.

3) The authrs use without clear rational and consistency Xenopus (the genus name), Xenopus laevis, X. laevis and X. tropicalis (species names). This is very confusing, especially since according to the title this is a review on Xenopus laevis. In addition, there is inconsistency in italicizing Xenopus and laevis or tropicalis. All species’ names should be italicized. After mentioning the Xenopus laevis fully, the abbreviation X. laevis should be used systematically in the whole text.  

We revised it.

4) Regarding the list of relevant contributions to scientific fields by X. laevis, the authors should consider adding immunology. Also, since this list is repeated several times in different sections, a table would be useful and prevent these repetitions.

As suggested now and later, we added a paragraph on immunology.

5) The English would merit to be improved over the whole document. There are multiple inaccuracies, redundancies, and grammatical mistakes.

English was revised.

Other minor points

1) Simple summary, line 10-11 is a bit awkward. Suggest: Among advantages of this species one can mention:… 

We modified the sentence.

2) Abstract line 19: it would be relevant to add immunology. 

As suggested now and later, we added a paragraph on immunology.

3) Abstract, sentence line 21-22: delete “still”. Also, it is unclear what the authors mean by “diffuse”. Zebrafish is a useful and powerful alternative model, so it doesn’t seem appropriate or productive to be condescending or pejorative. Please revise.

We modified the sentence.

4) Abstract, line 27: I don’t think that “higher genetic homology” is correct. Gene homology refers to an inference of common ancestor between two genes. This is not a metric. Suggesting changing by: “the X. laevis genome displays a higher degree of similarity with that of mammals.” 

Thanks for the suggestion. We modified it.

5) Introduction, line 35: African clawed frog (add clawed), line 36 “has been” (not was), line 38 “many” eggs

We revised it.

6) Line 42-43: the authors may want to consider adding immunotoxicology. Several studies in X. laevis have been published in mainstream scientific journals over the last 10 years.

As suggested now and later, we added a paragraph on immunology

7) Line 50: delete the end of the sentence (and therefore...) that is redundant.

Done

8) Line 82: Start a new sentence: “It was this ability…”

Done

9) Figure legend 2 should use specifically X. laevis (all in italic) not Xenopus.

Done

13) Line 113: Rana pipiens in italic

Revised

14) Line 118: X. laevis is fully resistant to Chytrid fungi (2 species), whereas X. tropicalis is susceptible. The authors should be accurate in their text. Other more relevant pathogens should also be mentioned including ranaviruses and mycobacteria (with references).

To avoid other references, we deleted the sentence.

15) Line 126-127: this is a useless repetition from the introduction. Important references on tumor models (e.g., several well characterized transplantable lymphoid tumors) are missing, and again the extensive contribution of X. laevis to developmental immunology and immuntoxicology should be considered.

As suggested now and later, we added a paragraph on immunology

16) Line 136-136: Again, the use of homology in this sentence should be defined in more details. What 79% homology of genome means? Is this related to the number of gene homologs or the extent of sequence identity?

We modified and completed the sentence in order to be more clear.

17) End of section 3. Most reference cited about cancer studies in X. laevis are reviews. The authors should cite primary references not just reviews. Besides studies in embryos, lymphoid tumors and anti-tumor responses have been extensiuvely characterized in X. laevis. This work cannot be omitted.

As suggested now and later, we added a paragraph on immunology (8.1). Moreover, we tried to reduce the number of citations by preferring research papers rather than reviews.

18) Section 5, line 228, one can perhaps also mention that tissue culture and live imaging for X. laevis can be done at room temperature (although optimal temperature for X. laevis cell lines is 27ºC), thus not necessitating special devices to warm up at 37ºC.

Revised. Thank you.

19) Sentence line 233- 234 regarding regeneration needs to be more specific and detailed. The regeneration capacity in X. laevis occurs from tailbud stage to pre-metamorphic stage 53 but is transiently lost from stage 45-47. Two critical recent work on this by J Gurdon group in Science 2019 and Development 2020, should be cited. 

Two new references were added.

20) Section 9, line 420 “some studies have been published. For example, Pekmezekmek… recently reported (not proved!). Also please split this sentence that is far too long and confusing.

We rephrased it.

21) Line 479-480: the authors should consider to also cite the drastic effects of atrazine and endocrine disruptor chemicals associated with hydrofraking on immunity in X. laevis

Thank you for the suggestion. We believe that the review is already rich enough in information. 

22) It should be noted that in first figure (Visual summary?) before the introduction the frog depicted is not X. laevis but X. tropicalis. Since this review is focused on X. laevis as shown in the title, it would be better to find a drawing or picture of X. laevis.

We wonder why reviewer 1 perceives it as X. tropicalis, not X. laevis. Maybe for the lighter coat? in any case, we are 100% sure that it is X. laevis, considering that the drawing was done by us, in order to avoid copyrighted images. To avoid misunderstanding, we changed the coat color.

Reviewer 2 Report

The paper discusses Xenopus laevis, a type of anuran amphibian, as a reliable model organism in various bioscience fields. It argues that the organism has been widely used to study embryonic development because of its evolutionary closeness to higher vertebrates in terms of physiology, gene expression, and organ development. The paper appears to be logically organized and covers a comprehensive range of relevant topics on Xenopus laevis. References are extensively cited.   It is a nice review paper in my view though I just have a couple of minor suggestions for the paper. None of my comment will sink the paper.   1- Line 514: a typo here. This should be section 11. 2- Line 521: "key" -> "key words"? 3- Maybe this is a subject comment, please feel free to accept this or no. the sections seem to move between different fields of study where Xenopus laevis is used, then to environmental issues, then back to bioscience. It might be more logical to keep similar topics grouped together. For example, group all sections related to developmental biology (oogenesis, organogenesis, apoptosis) together, then move to other topics like environmental studies and genetic disease studies. A suggested structure could be as follow, but please note this is just a reference.
  1. Introduction
  2. The advantages of the Xenopus model
  3. Xenopus in laboratory
  4. Xenopus in Developmental Studies:
    • Xenopus in the study of oogenesis and role of cytoskeletal proteins
    • Xenopus as a model for the study of organogenesis
    • Interactions of specific proteins in organ formation: the eye and kidney
    • Xenopus and studies of apoptosis in development
  5. Xenopus in Environmental Studies:
    • Microplastic environmental pollution
    • Xenopus in the studies of embryo toxicology
  6. Xenopus in Genetic and Disease Studies:
    • Xenopus laevis to study and model human genetic disease
  7. Xenopus in Emerging Areas of Bioscience:
    • Nanoscience and Xenopus developmental studies
  8. Conclusions

Author Response

We thank you for this suggestion, which we gladly accepted. Now the review follows the structure you suggested.

Reviewer 3 Report

I do not like your current paper title (see my comments) because it does not reflect your content.

Xenopus laevis is a fascinating model organism in the historical context (e.g., pregnancy tests), established research areas (e.g., Evo-Devo), and newer research areas such as nanoparticles or toxicology. I have provided several corrections, suggestions, and comments on the PDF and hope they are helpful for your revision. All scientific names must be italicized including the genus and the species epithet. I did not understand several of your statements (see my comments).  The first figure lacks a figure number and legend. Explain abbreviations when they first appear. Avoid repetitions, currently there are many (see my comments). The graphic showing published papers about Xenopus and toxicology is very good, but have the older years on the left and younger on the right. Consider having a similar figure for Xenopus and its other research areas. The best way to evaluate Xenopus laevis as a model is if you present published papers for other vertebrate lab models (e.g., zebrafish, axolotl, mouse) and toxicology. I am sure the reader would love to see where Xenopus stands in comparison with other model organism.

I provided some corrections in the PDF and marked statements I did not understand. Keep in mind that sentences in the English language are generally short; avoid long multi-cause sentences. 

Author Response

Thanks for the thorough review. You will find our answers in the pdf.

Round 2

Reviewer 3 Report

You have carefully revised the manuscript and reads much better than before. Minor errors include spacing problems and inconsistencies regarding the use of italics for scientific names (zebrafish should not be in italics). I did not find a rebuttal letter or your response to my previous comments/suggestions. The image shown in front of the introduction nicely reflects the use of X. laevis as a model organism, and the image should be referred to in your paper with a figure number and legend. Here it is misused as decoration which I find unacceptable in the sciences.

Author Response

Dear Reviewer, 

We have responded to the previous reviews point by point, replying to every comment in the PDF file you attached and reloading it with our responses to your comments. To be fair, we upload it again,
you will find it as an attachment.  In that PDF file, we also explained that the image in front of the introduction is actually a Graphical Abstract, which therefore doesn't request legend or figure number.
The Graphical Abstract is mandatory according to the rules of the journal, and it has to be inserted immediately after the abstract. This is why, for the second time, you don't find that figure with number and legend: it is not a decoration, it is just a Graphical Abstract. 
For this second round, we reviewed the other small errors and we warmly thank you for the revisions. 
